# SMPLOlympics: Sports Environments for Physically Simulated Humanoids

https://SMPLOlympics.github.io/

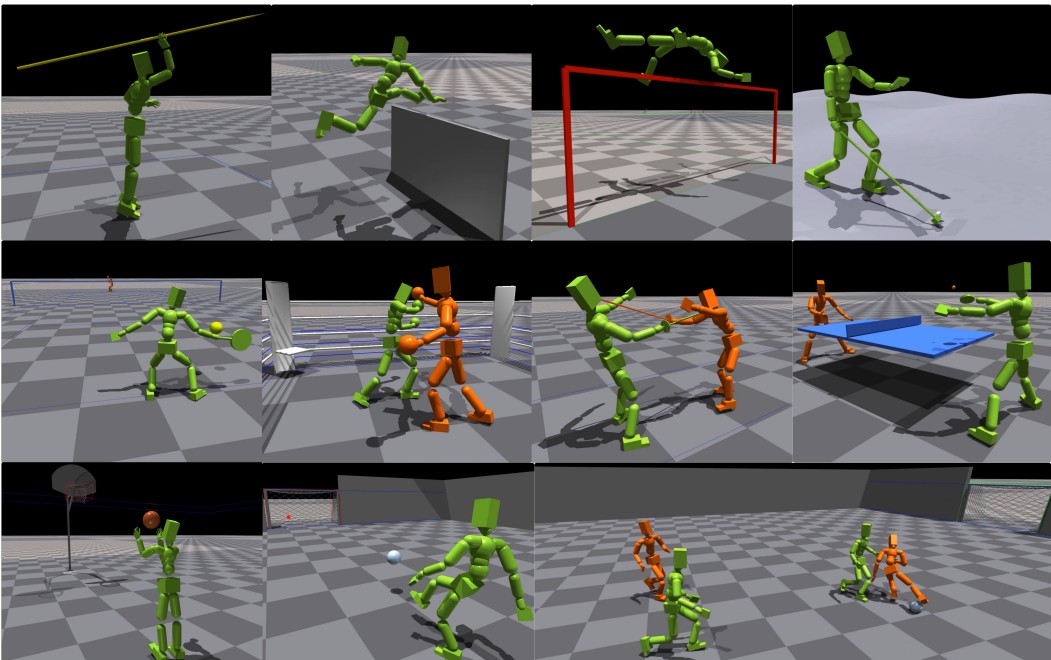

Figure 1: A collection of various sports environments for physically simulated humanoids.

## Abstract

We present SMPLOlympics, a collection of physically simulated environments that allow humanoids to compete in a variety of Olympic sports. Sports simulation offers a rich and standardized testing ground for evaluating and improving the capabilities of learning algorithms due to the diversity and physically demanding nature of athletic activities. As humans have been competing in these sports for many years, there is also a plethora of existing knowledge on the preferred strategy to achieve better performance. To leverage these existing human demonstrations from videos and motion capture, we design our humanoid to be compatible with the widely-used SMPL and SMPL-X human models from the vision and graphics community. We provide a suite of individual sports environments, including golf, javelin throw, high jump, long jump, and hurdling, as well as competitive sports, including both 1v1 and 2v2 games such as table tennis, tennis, fencing, boxing, soccer, and basketball. Our analysis shows that combining strong motion priors with simple rewards can result in human-like behavior in various sports. By providing a unified sports benchmark and baseline implementation of state and reward designs, we hope that SMPLOlympics can help the control and animation communities achieve human-like and performant behaviors.

Submitted to the 38th Conference on Neural Information Processing Systems (NeurIPS 2024) Track on Datasets and Benchmarks. Do not distribute.

# 1 Introduction

Competitive sports, much like their role in human society, offer a standardized way of measuring the performance of learning algorithms and creating emergent human behavior. While there exist isolated efforts to bring individual sport into physics simulation [8, 34, 7, 33, 27], each work uses a different humanoid, simulator, and learning algorithm, which prevents unified evaluation. Their specially built humanoids also make it difficult to acquire compatible motion data, as retargeting might be required to translate motion to each humanoid. Building a collection of simulated sports environments that uses a shared humanoid embodiment and training pipeline is challenging, as it requires expert knowledge in humanoid design, reinforcement learning (RL), and physics simulation.

These challenges have led to previous benchmarks and simulated environments [3, 25] focusing mainly on locomotion tasks for humanoids. While these tasks (e.g., moving forward, getting up from the ground, traversing terrains) are as benchmarks, they lack the depth and diversity needed to induce a wide range of behaviors and strategies. As a result, these environments do not fully exploit the potential of humanoids to discover actions and skills found in real-world human activities.

Another important aspect of working with simulated humanoids is the ease of obtaining human demonstrations. The resemblance to the human body makes humanoids capable of performing a diverse set of skills; a human can also easily judge the strategies used by humanoids. Curated human motion can be used either as motion prior [17, 18, 24] or in evaluation protocols. Thus, having an easy way to obtain new human motion data compatible with the humanoid, either from motion capture (MoCap) or videos, is critical for simulated humanoid environments.

In this work, we propose SMPLOlympics, a collection of physically simulated environments for a variety of Olympic sports. SMPLOlympics offers a wide range of sports scenarios that require not only locomotion skills, but also manipulation, coordination, and planning. Unified under one humanoid embodiment, our environments provide a rich set of challenges for developing and testing embodied agents. We use humanoids compatible with the SMPL family of models, which enables the direct conversion of human motion in the SMPL format to our humanoid. For tasks that require articulated fingers, we use SMPL-X [16] based humanoid which has a much higher degree of freedom (DOF); for tasks that do not need hands, we use SMPL [2]. As popular human models, the SMPL family of models is widely adopted in the vision and graphics community, which provides us with access to human pose estimation methods [32] capable of extracting coherent motion from videos. The existing large-scale human motion dataset [13] in the SMPL format also helps build general-purpose motion representation for humanoids [10].

Our sports environments support both individual and competitive sports, providing a comprehensive platform for testing and benchmarking. For individual sports, we include activities such as golf, javelin throw, high jump, long jump, and hurdling. Competitive sports in our suite include 1v1 games such as ping pong, tennis, fencing, and boxing, as well as team sports such as soccer and basketball. To facilitate benchmarking, we also include tasks such as penalty kicks (for soccer) and ball-target hitting (for ping-pong and tennis) that are easy to measure performance. To demonstrate the importance of human demonstrations, we extract motion from videos using off-the-shelf pose estimation methods, and show that using human motion data as motion prior can [18] significantly improves human likeness in the resulting motion. We also test recent motion representations in simulated humanoid control using hierarchical RL [10], and show that a learned motion representation combined with simple rewards can lead to many versatile human-like behaviors to achieve impressive sports results (*i.e.* discovering the Fosbury way for high jump).

In conclusion, our contributions are: (1) we propose SMPLOlympics, a collection of simulated environments that allow humanoids to compete in a variety of Olympic sports; (2) we extract human demonstration data from videos and show their effectiveness in helping build human-like strategies in simulated sports; (3) we provide the starting state and reward designs for each sport, benchmark state-of-the-art algorithms, and show that simple rewards combined with a strong motion prior can lead to impressive sports feats.

## 2  Related Works

**Simulated Humanoid Sports**. Simulated humanoid sports can help generate animations and explore optimal sports strategies. Research has focused on various individual sports within simulated environments, including tennis [34], boxing [27, 36], fencing [27], basketball dribbling [7] and soccer [29, 8]. These studies leverage human motion to achieve human-like behaviors, using it to acquire motor skills [8, 27] or establish motion prior [34]. However, the diversity in humanoid definitions across studies makes it difficult to aggregate additional human demonstration data due to the need for retargetting. Furthermore, the task-specific training pipelines in these studies are hard to generalize to new sports. In contrast, SMPLOlympics provides a unified benchmark employing a consistent humanoid model and training pipeline across all sports. This standardization not only facilitates the extension to more sports, but also simplifies the process of benchmarking learning algorithms.

**Simulated RL Benchmarks**. Simulated full-body humanoids provide a valuable platform for studying embodied intelligence due to their close resemblance to real-world human behavior and physical interactions. Current RL benchmarks [3, 25, 14] often focus on locomotion tasks such as moving forward and traversing terrain. `dm_control` [25] and OpenAI [3] Gym only include locomotion tasks. ASE [19] includes results for five tasks based on mocap data, which involve mainly simple locomotion and sword-swinging actions. These tasks lack the complexity required to fully exploit the capabilities of simulated humanoids. Sports scenarios require agile motion and strategic teamwork. They are also easily interpretable and provide measurable outcomes for success. A concurrent work, HumanoidBench [23] employs a commercially available humanoid robot in simulation to address 27 locomotion and manipulation tasks. Unlike HumanoidBench, ours targets competitive sports and uses available human demonstration data to enhance the learning of human-like behaviors. This emphasis is essential, as without human demonstrations, behaviors developed in benchmarks can often appear erratic, nonhuman-like, and inefficient.

**Humanoid Motion Representation**. Adversarial learning has proven to be a powerful method for using human reference motions to enhance the naturalness of humanoid animations [18, 30, 1]. Due to the high DoF in humanoids and the inherent sample inefficiency of RL training, efforts have focused on developing motion primitives [6, 15, 5, 20] and motion latent spaces [4, 19, 24]. These techniques aim to accelerate training and provide human-like motion priors. Notably, approaches such as ASE [19], CASE [4], and CALM [24] utilize adversarial learning objectives to encourage mapping between random noise and realistic motor behavior. Furthermore, methods such as ControlVAE [31], NPMP [15], PhysicsVAE [28], NCP [36], and PULSE [10] leverage the motion imitation task to acquire and reuse motor skills for the learning of downstream tasks. In this work, we study AMP [18] and PULSE [10] as exemplary methods to provide motion priors. Our findings demonstrate that a robust motion prior, combined with straightforward reward designs, can effectively induce human-like behaviors in solving complex sports tasks.

## 3  Preliminaries

We define the full-body human pose as $\boldsymbol{q}_t \triangleq (\boldsymbol{\theta}_t, \boldsymbol{p}_t)$, consisting of 3D joint rotations $\boldsymbol{\theta}_t \in \mathbb{R}^{J \times 6}$ and positions $\boldsymbol{p}_t \in \mathbb{R}^{J \times 3}$ of all $J$ joints on the humanoid, using the 6 DoF rotation representation [35]. To define velocities $\dot{\boldsymbol{q}}_{1:T}$, we have $\dot{\boldsymbol{q}}_t \triangleq (\boldsymbol{\omega}_t, \boldsymbol{v}_t)$ as angular $\boldsymbol{\omega}_t \in \mathbb{R}^{J \times 3}$ and linear velocities $\boldsymbol{v}_t \in \mathbb{R}^{J \times 3}$. If an object is involved (*e.g.* javelin, football, ping-pong ball), we define their 3D trajectories $\boldsymbol{q}_t^{\text{obj}}$ using object position $\boldsymbol{p}_t^{\text{obj}}$, orientation $\boldsymbol{\theta}_t^{\text{obj}}$, linear velocity $\boldsymbol{v}_t^{\text{obj}}$, and angular velocity $\boldsymbol{\omega}_t^{\text{obj}}$. As a notation convention, we use $\widehat{\cdot}$ to denote the ground truth kinematic quantities from Motion Capture (MoCap) and normal symbols without accents for values from the physics simulation.

**Goal-conditioned Reinforcement Learning for Humanoid Control**. We define each sport using the general framework of goal-conditioned RL. Namely, a goal-conditioned policy $\pi_{\text{task}}$ is trained to control a simulated humanoid competing in a sports environment. The learning task is formulated as a Markov Decision Process (MDP) defined by the tuple $\mathcal{M} = \langle \boldsymbol{S}, \boldsymbol{A}, \boldsymbol{T}, \boldsymbol{\mathcal{R}}, \gamma \rangle$ of states, actions,

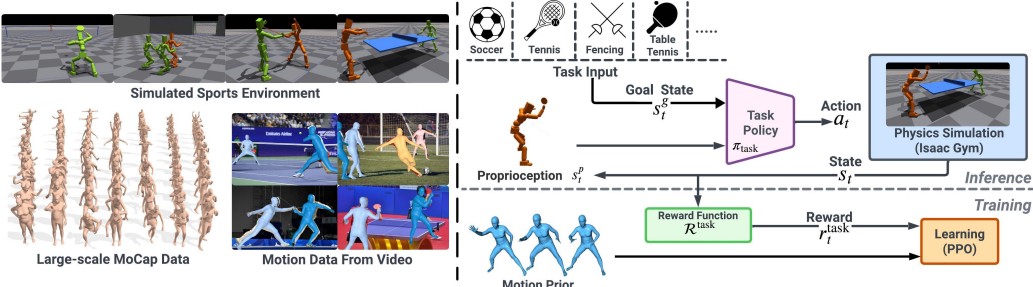

Figure 2: An overview of SMPLOlympics: we design a collection of simulated sports environments and leverage RL and human demonstrations (from videos or MoCap) as prior to tackle them.

116 transition dynamics, reward function, and discount factor. The simulation determines the state
117 $s_t \in S$ and transition dynamics $T$, where a policy computes the action $a_t$. The state $s_t$ contains the
118 proprioception $s_t^p$ and the goal state $s_t^g$. Proprioception is defined as $s_t^p \triangleq (q_t, \dot{q}_t)$, which contains
119 the 3D body pose $q_t$ and velocity $\dot{q}_t$. We use $b$ to indicate the boundary of the arena to which a sport
120 is limited. All values are normalized with respect to the humanoid heading (yaw).

# 4 SMPLOlympics: sports environments For Simulated Humanoids

122 In this section, we describe the formulation of each of our sports environments, from single-person
123 sports (Sec. 4.1) to multi-person sports (Sec. 4.2). Then, we describe our pipeline for acquiring
124 human demonstration data from videos (Sec. 4.3). An overview can be found in Fig. 2. For each
125 sport, we provide a preliminary reward design that serves as a baseline for future research. Due to
126 space constraints, omitted details can be found in the supplement.

## 4.1 Single-person Sports

128 **High Jump**. In the high jump environment, the humanoid's objective is to jump over a horizontal
129 bar placed at a certain height without touching it. The bar is positioned following the setup of the
130 official Olympic game. The high jump goal state $s_t^{\text{g-high\_jump}} = (p_t^b, p_t^l)$ contains the positions of the
131 bar $p_t^b \in \mathbb{R}^3$ and the landing area $p_t^l \in \mathbb{R}^3$. The reward is defined as $\mathcal{R}^{\text{high jump}}(s_t^p, s_t^{\text{g-high\_jump}}) =$
132 $w^p r_t^p + w^h r_t^h$. The position reward $r_t^p$ encourages the humanoid to go closer to the goal point, which
133 is behind the high jump bar. The height reward $r_t^h$ encourages the humanoid to jump higher. Training
134 terminates when the humanoid is in contact with the bar, does not pass the bar, or falls to the ground
135 before jumping. We also set up four bar heights for curriculum learning: 0.5m, 1m, 1.5m, and 2m.

136 **Long Jump**. Long jump is also set similar to the Olympic games, with a 20m runway followed
137 by a jump area. Before the humanoid jumps, its feet should be behind the jump line. The goal
138 state $s_t^{\text{g-long\_jump}} \triangleq (p_t^s, p_t^l, p_t^g)$ includes the position of the starting point $p_t^s \in \mathbb{R}^3$, jump line
139 $p_t^l \in \mathbb{R}^3$, and the goal $p_t^g \in \mathbb{R}^3$. The training reward is defined as $\mathcal{R}^{\text{long jump}}(s_t^p, s_t^{\text{g-long\_jump}}) \triangleq$
140 $w^p r_t^p + w^v r_t^v + w^h r_t^h + w^l r_t^l$. The position reward $r_t^p$ encourages the humanoid to get closer to the
141 goal, the velocity reward $r_t^v$ encourages larger running speed, and the height reward $r_t^h$ encourages
142 higher jump. Finally, $r_t^l$ encourages jumping far.

143 **Hurdling**. In hurdling, the humanoid tries to reach a finishing line 110 meters ahead and needs to
144 jump over 10 hurdles (each 1.067m high, placed 13.72m from the start, with subsequent hurdles
145 spaced every 9.14m). The goal state is defined as $s_t^{\text{g-hurdling}} \triangleq (p_t^h, p_t^f)$, where $p_t^h \in \mathbb{R}^{10 \times 3}$ and
146 $p_t^f \in \mathbb{R}^3$ includes the positions of these hurdles as well as the finish line. We define a simple reward
147 function as $\mathcal{R}^{\text{hurdling}}(s_t^p, s_t^{\text{g-hurdling}}) = r_t^{\text{distance}}$. $\mathcal{R}^{\text{hurdling}}$ encourages the agent to run towards the finish
148 line and clear each hurdle. Additionally, we employ a curriculum for hurdling, where the height of
149 each hurdle is randomly sampled between 0 and 1.167 meters for each episode.

150 **Golf**. For golf, the humanoid's right hand is replaced with a golf club measuring 1.14 meters. The
151 driver of the golf club is simulated as a small box ( 0.05m × 0.025m × 0.02m). We incorporate a

randomly generated terrain in the golf environment, designed to mimic real-world grasslands with wave-like features and an amplitude of 0.5 meters. The objective for the humanoid is to hit the ball towards a randomly sampled target position. The goal state $s_t^{\text{g-golf}} \triangleq (p_t^b, p_t^c, p_t^g, o_t)$ includes the ball position $p_t^b \in \mathbb{R}^3$, club $c_t^b \in \mathbb{R}^3$, goal position $p_t^g \in \mathbb{R}^3$, and terrain height map $o_t \in \mathbb{R}^{32 \times 32}$. The reward is defined as $\mathcal{R}^{\text{golf}}(s_t^{\text{p}}, s_t^{\text{g-golf}}) = w^{\text{p}} r_t^{\text{p}} + w^{\text{c}} r_t^{\text{c}} + w^{\text{g}} r_t^{\text{g}} + w^{\text{pred}} r_t^{\text{pred}}$, where the $r_t^{\text{p}}$ encourages the ball to move forward, $r_t^{\text{c}}$ encourages swinging the golf club to hit the ball, and $r_t^{\text{g}}$ encourages the ball to reach the goal. In addition, we predict the ball's trajectory and provide a dense reward $r_t^{\text{pred}}$ based on the distance between the predicted landing point and the goal.

**Javelin**. For javelin throw, we use SMPL-X humanoid with articulated fingers. The goal state is defined as $s_t^{\text{g-javelin}} \triangleq (q_t^{\text{obj}}, p_t^r, p_t^h)$, where $q_t^{\text{obj}} \in \mathbb{R}^{13}$, includes the position, orientation, linear, and angular velocity of the javelin. $p_t^r$ and $p_t^h$ are the positions of the root and right hand. The reward is defined as $\mathcal{R}^{\text{javelin}}(s_t^{\text{p}}, s_t^{\text{g-javelin}}) \triangleq w^{\text{grab}} r_t^{\text{grab}} + w^{\text{js}} r_t^{\text{js}} + w^{\text{goal}} r_t^{\text{goal}} + w^{\text{s}} r_t^{\text{s}}$. The grab reward $r_t^{\text{grab}}$ encourages the right hand to grab the javelin. The javelin stability reward $r_t^{\text{js}}$ minimizes the javelin's self-rotation. The goal reward $r_t^{\text{goal}}$ encourages the humanoid to throw the javelin further. The stability reward $r_t^{\text{s}}$ is to avoid large movements of the body.

## 4.2 Multi-person Sports

**Tennis**. For tennis, each humanoid's right hand is replaced as an oval racket. We use the same measurement as a real tennis court and ball. We design two tasks: a single-player task where the humanoid trains to hit balls launched randomly, and a 1v1 mode where the humanoid plays against another humanoid. For both tasks, the goal state is defined as $s_t^{\text{g-tennis}} \triangleq (p_t^{\text{ball}}, v_t^{\text{ball}}, p_t^{\text{racket}}, p_t^{\text{tar}}$, where $p_t^{\text{ball}} \in \mathbb{R}^3, v_t^{\text{ball}} \in \mathbb{R}^3, p_t^{\text{racket}} \in \mathbb{R}^3$ and $p_t^{\text{tar}} \in \mathbb{R}^3$, which includes the position and velocity of the ball, position of the racket and position of the target. The reward function for tennis is defined as $\mathcal{R}^{\text{tennis}}(s_t^{\text{p}}, s_t^{\text{g-tennis}}) = w_{\text{p}} r_t^{\text{racket}} + w_{\text{b}} r_t^{\text{ball}}$. The racket reward $r_t^{\text{racket}}$ encourages the racket to reach the ball, and the ball reward $r_t^{\text{ball}}$ aims to successfully hit the ball into the opponent's court, as close to the target as possible. For the single-player task, we shoot a ball from the opposite side from a random position and trajectory, simulating a ball hit by the opponent. The target $p_t^{\text{tar}}$ is also randomly sampled. For the 1v1 scenario, we can either train models from scratch or initialize two identical single-player models as opponents, which can play back and forth.

**Table Tennis**. For table tennis, each humanoid is equipped with a circular paddle (replacing the right hand) and play on a standard table. Similar to tennis, we have the single-player task and the 1v1 task. Similarly, the goal state is defined as $s_t^{\text{g-tennis}} \triangleq (p_t^{\text{ball}}, v_t^{\text{ball}}, p_t^{\text{racket}}, p_t^{\text{tar}})$. The reward function for table tennis is defined as $\mathcal{R}^{\text{table tennis}}(s_t^{\text{p}}, s_t^{\text{g-table\_tennis}}) = w_{\text{p}} r_t^{\text{racket}} + w_{\text{b}} r_t^{\text{ball}}$. The paddle reward $r_t^{\text{racket}}$ is the same as the tennis while we modify the $r_t^{\text{ball}}$ slightly to encourage more hits for table tennis.

**Fencing**. For 1v1 fencing, each humanoid is equipped with a sword (replacing the right hand) and plays on a standard fencing field. The goal state is defined as $s_t^{\text{g-fencing}} \triangleq (p_t^{\text{opp}}, v_t^{\text{opp}}, p_t^{\text{sword}} - p_t^{\text{opp-target}}, \|c_t\|_2^2, \|c_t^{\text{opp}}\|_2^2, b)$, which contains the opponent's position body $p_t^{\text{opp}} \in \mathbb{R}^{24 \times 3}$, linear velocity $v_t^{\text{opp}} \in \mathbb{R}^{24 \times 3}$, the difference between target body position $p_t^{\text{opp-target}} \in \mathbb{R}^{5 \times 3}$ on the opponent and agent's sword tip position $p_t^{\text{sword}}$, normalized contract forces on the agent itself $\|c_t\|_2^2 \in \mathbb{R}^{24 \times 3}$ and its opponent $\|c_t^{\text{opp}}\|_2^2 \in \mathbb{R}^{24 \times 3}$, as well as the bounding box $b \in \mathbb{R}^4$. To train the fencing agent, we define the fencing reward function as $\mathcal{R}^{\text{fencing}}(s_t^{\text{p}}, s_t^{\text{g-fencing}}) = w_{\text{f}} r_t^{\text{facing}} + w_{\text{v}} r_t^{\text{vel}} + w_{\text{s}} r_t^{\text{strike}} + w_{\text{p}} r_t^{\text{point}}$. The facing $r_t^{\text{facing}}$ and velocity reward $r_t^{\text{vel}}$ encourage the agent to face and move toward the opponent. The strike reward $r_t^{\text{strike}}$ encourages the agent's sword tip to get close to the target, while $r_t^{\text{point}}$ is the reward for getting in contact with the target. We use the pelvis, head, spine, chest, and torso as the target bodies. The episode terminates if either of the humanoids falls or steps out of bounds.

**Boxing**. For boxing, we simulate two humanoids with sphere hands in a bounded arena. The goal state is similar to fencing: $s_t^{\text{g-boxing}} \triangleq (p_t^{\text{opp}}, v_t^{\text{opp}}, p_t^{\text{hand}} - p_t^{\text{opp-target}}, \|c_t\|_2^2, \|c_t^{\text{opp}}\|_2^2)$ but without the bounding box information. The reward function and target body parts are also the same as fencing, though replacing the sword tip to the hands.

**Soccer**. The soccer environment includes one or more humanoids, a ball, two goal posts, and the field boundaries. The field measures 32m × 20m. We support three tasks: penalty kicks, 1v1, and 2v2.

For penalty kicks, the humanoid is positioned 13 meters from the goal line, with the ball placed at a fixed spot 12 meters directly in front of the goal center. The objective is to kick the ball toward a randomly sampled target within the goal post. To achieve this, the controller is provided $s_t^{\text{g-kick}} \triangleq (p_t^{\text{ball}}, \dot{q}_t^{\text{ball}}, p_t^{\text{goal-post}}, p_t^{\text{goal-target}})$, where $p_t^{\text{ball}} \in \mathbb{R}^3$ is the ball position, $\dot{q}_t^{\text{ball}} \in \mathbb{R}^3$ is the velocity and angular velocity, $p_t^{\text{goal-post}} \in \mathbb{R}^4$ is the bounding box of the goal, and $p_t^{\text{goal-target}} \in \mathbb{R}^3$ is the target location within the goal post. The reward is $\mathcal{R}^{\text{soccer-kick}}(s_t^{\text{p}}, s_t^{\text{g-kick}}) \triangleq w^{\text{p2b}}r^{\text{p2b}} + w^{\text{b2g}}r^{\text{b2g}} + w^{\text{bv2g}}r^{\text{bv2g}} + w^{\text{b2t}}r^{\text{b2t}} - c_t^{\text{no-dribble}}$. Various rewards are designed to guide the character towards a run-and-kick motion. The player-to-ball ($r^{\text{p2b}}$) reward motivates the character to move towards the ball. The ball-to-goal reward ($r^{\text{b2g}}$) reduces the distance between the ball and the target. The ball-velocity-to-goal ($r^{\text{bv2g}}$) encourages a higher velocity of the ball toward the target. The ball-to-target ($r^{\text{b2t}}$) reward encourages a smaller distance between the target and the predicted landing spot of the ball based on its current position and velocity. Finally, a negative reward ($c_t^{\text{no-dribble}}$) is applied if the character passes the spawn position of the ball, which discourages dribbling and encourages kicking.

Beyond penalty kicks, we explore team-play dynamics, including 1v1 and 2v2. The controller is provided with a state defined as $s_t^{\text{g-soccer}} \triangleq (p_t^{\text{ball}}, \dot{q}_t^{\text{ball}}, p_t^{\text{goal-post}}, p_t^{\text{ally-root}}, p_t^{\text{opp-root}})$, where $p_t^{\text{ally-root}} \in \mathbb{R}^3$ and $p_t^{\text{opp-root}} \in \mathbb{R}^3$ are the root positions of the ally and opponents (1 or 2). The controller is then trained using the following reward $\mathcal{R}^{\text{soccer-match}}(s_t^{\text{p}}, s_t^{\text{g-soccer}}) \triangleq w^{\text{p2b}}r^{\text{p2b}} + w^{\text{b2g}}r^{\text{b2g}} + w^{\text{bv2g}}r^{\text{bv2g}} + w^{\text{point}}r^{\text{point}}$, where $r^{\text{p2b}}$, $r^{\text{b2g}}$ and $r^{\text{bv2g}}$ are the same as in penalty kick. $r^{\text{b2g}}$ and $r^{\text{bv2g}}$ are zeroed out when the distance to the ball is greater than 0.5m. $r^{\text{point}}$, the scoring a goal, provides a one-time bonus and or penalty for goals. Notice that this is a rudimentary reward design compared to prior art [8] and serves as a starting point for further development.

**Basketball**. Our basketball environment is set up similarly to the soccer environment except for using the SMPL-X humanoid. The court measures 29m × 15m, with a 3m high hoop. We also introduce the task of free-throw, where the humanoid begins at a distance of 4.5 meters from the hoop with the ball initially positioned close to its hands. The objective is to successfully throw the basketball into the hoop. The goal state for this task is defined similarly to that of the soccer penalty kicks, with the distinction being the prohibition of foot-to-ball contact to maintain basketball rules.

**Competitive Self-play**. In competitive sports environments, we implement a basic adversarial self-play mechanism where two policies, initialized randomly, compete against each other to optimize their rewards. We adopt an alternating optimization strategy from [27], where one policy is frozen while the other is trained. This encourages each policy to develop offensive and defensive strategies, contributing to more competitive behavior, as observed in boxing and fencing (`supplement site`).

### 4.3 Acquiring Human Demonstration From Videos

We utilize TRAM [26] for 3D motion reconstruction from Internet videos, providing robust global trajectory and pose estimation under dynamic camera movements, commonly found in sports broadcasting. Specifically, TRAM estimates SMPL parameters [9] which include global root translation, orientation, body poses, and shape parameters. We further apply PHC [11], a physics-based motion tracker, to imitate these estimated motions, ensuring physical plausibility. We find these corrected motions are significantly more effective as positive samples for adversarial learning compared to raw estimated results. More details and ablation are provided in the supplementary materials.

## 5 Experiments

**Implementation Details**. Simulation is conducted in Isaac Gym [14], where the policy runs at 30 Hz and the simulation at 60 Hz. All task policies utilize three-layer MLPs with units [2048, 1024, 512]. The SMPL humanoid models adhere to the SMPL kinematic structure, featuring 24 joints, 23 of which are actuated, yielding an action space of $\mathcal{R}^{69}$. The SMPL-X humanoid has 52 joints,

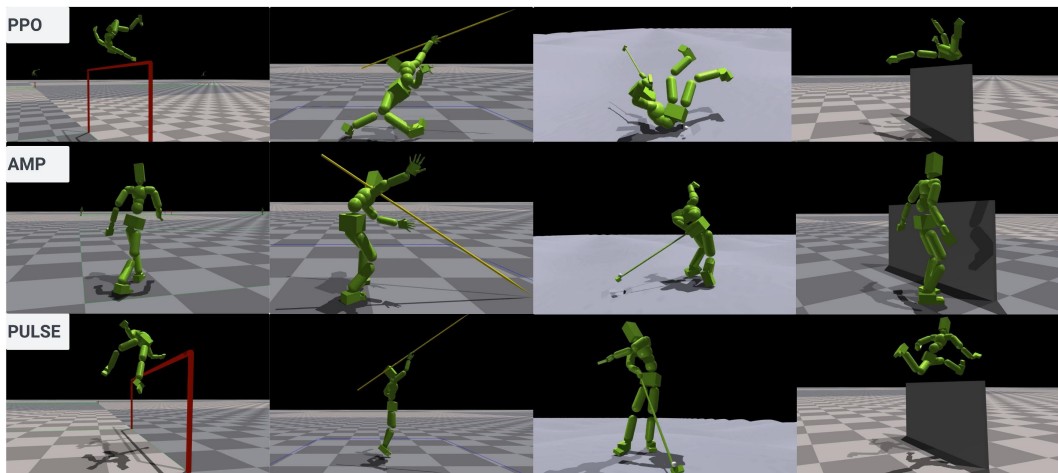

Figure 3: Qualitative results for high jump, javelin, golf, and hurdling. PPO and AMP try to solve the task using inhuman behavior, while PULSE can discover human-like behavior.

51 actuated, including 21 body joints and hands, resulting in an action space of $\mathcal{R}^{153}$. Body parts on our humanoid consist of primitives such as capsules and blocks. All models can be trained on a single Nvidia RTX 3090 GPU in 1-3 days. We limit all joint actuation forces to 500 Nm. For more implementation details, please refer to the supplement.

**Baselines**. We benchmark our simulated sports using some of the state-of-the-art simulated humanoid control methods. While not a comprehensive list, it provides a baseline for the challenging environments. Each task is trained using PPO [22], AMP [18], PULSE [10], and a combination of PULSE and AMP. AMP use a discriminator with the policy to provide an adversarial reward, using human demonstration data to deliver a "style" reward that reflects the human-likeness of humanoid motion. Both task and discriminator rewards are equally weighted at 0.5. PULSE extracts a 32-dimensional universal motion representation from AMASS data, surpassing previous methods [24, 19] in coverage of motor skills and applicability to downstream tasks. Compared to AMP, PULSE uses hierarchical RL and offers a learned action space that accelerates training and provides human-like motion prior (instead of a discriminative reward). PULSE and AMP can be combined effectively, where PULSE provides the action space and AMP provides task-specific style reward.

**Metrics**. We provide quantitative evaluations for tasks with easily measurable metrics such as high jump, long jump, hurdling, javelin, golf, single-player tennis, table tennis, penalty kicks, and free throws. These metrics are detailed in the supplementary materials, where we also present qualitative assessments for tasks that are more challenging to quantify, such as boxing, fencing, and team soccer. Specifically, success rate (Suc Rate) determines whether an agent completes a sport according to set rules. Average distance (Avg Dis) indicates the extent an agent or object travels. For sports involving ball hits, such as tennis and table tennis, we record the average number of successful ball strikes (Avg Hits). Error distance (Error Dis) measures the distance between the intended target and the actual landing spot, applicable in sports like golf, tennis, and penalty kicks. Additionally, the hit rate in golf quantifies the success of striking the ball with the club. Evaluations are performed on 1000 trials.

## 5.1 Benchmarking Popular Simulated Humanoid Algorithms

In this section, we evaluate the performance of various control methods across our sports environments. We provide qualitative results in Fig. 3 and Fig. 4, and training curves in Fig. 5. To view extensive qualitative results, including human-like soccer kick, boxing, high jump, *etc.*, please see `supplement`.

**Track & Field Sports (Without Video Data)**. We first evaluate track and field sports, including long jump, high jump, hurdling, and javelin throwing. For these sports, SOTA pose estimation methods fail to estimate coherent motion and global root trajectory as players and cameras are both fast-moving. Thus, we utilize a subset of the AMASS dataset containing locomotion data [21] as

Table 1: Evaluation on Long Jump, High Jump, Hurdling and Javelin. World records are in parentheses.

| | Long Jump (8.95m) | | High Jump (2.45m) | | | | Hurdling (12.8s) | | | Javelin (104.8m) | |
|---|---|---|---|---|---|---|---|---|---|---|---|
| Method | Suc Rate ↑ | Avg Dis ↑ | Suc Rate (1m) ↑ | Height (1m) ↑ | Suc Rate (1.5m) ↑ | Height (1.5m) ↑ | Suc Rate ↑ | Avg Dis ↑ | Time ↓ | Suc Rate ↑ | Avg Dis ↑ |
| PPO [22] | 53.6% | **19.42** | **100%** | **4.08** | **100%** | **4.11** | 57.6% | 108.9 | **11.22** | **100%** | **44.5** |
| AMP [18] | 0% | - | 0% | - | 0% | - | 0% | 13.24 | - | 0.31% | 2.03 |
| PULSE [10] | **100%** | 5.105 | **100%** | 2.01 | **100%** | 1.98 | **100%** | **122.1** | 17.76 | **100%** | 9.63 |

Table 2: Evaluation on Golf, Tennis, Table Tennis, Penalty Kick and Free Throw

| | Golf | | Tennis | | Table Tennis | | Penalty Kick | | Free Throw |
|---|---|---|---|---|---|---|---|---|---|
| Method | Hit Rate ↑ | Error Dis ↓ | Avg Hits ↑ | Error Dis ↓ | Avg Hits ↑ | Error Dis ↓ | Suc Rate ↑ | Error Dis ↓ | Suc Rate ↑ |
| PPO [22] | 0% | - | 2.76 | **1.92** | 1.01 | **0.06** | 0.0% | - | 0.0% |
| AMP [18] | **100%** | 1.43 | **3.95** | 5.30 | 1.10 | 0.13 | 0.0% | - | 0.0% |
| PULSE [10] | 99.9% | **1.29** | 2.48 | 3.50 | 0.74 | 0.19 | **76.6%** | **0.25** | **87.5%** |
| PULSE [10] + AMP [18] | 99.9% | 2.18 | 2.62 | 3.64 | **1.83** | 0.23 | 27.5% | 0.27 | 30.6% |

reference motions. Since PULSE is pretrained on AMASS, we exclude PULSE + AMP from these tests. Table 1 summarizes the quantitative results of different methods. In long jump, AMP fails entirely, often walking slowly to the jump line without a forward leap. This failure occurs because the policy prioritizes discriminator rewards over task completion. If the task is too hard, the policy will use simple motion (such as standing still) to maximize the discriminator reward instead of trying to complete the task. In contrast, PPO, while capable of jumping great distances, exhibits unnatural motions. PULSE successfully executes jumps with human-like motion, but lacks the specialized skills for top-tier records due to the absence of corresponding motion data in AMASS. The high jump displays similar patterns: PPO achieves impressive heights but with unnatural movements while AMP struggles to reconcile adversarial and task rewards. Surprisingly, as shown in Figure 3, PULSE successfully adopts a Fosbury flop approach without specific rewards to encourage this technique, likely leveraging breakdance skills. For hurdling, AMP completely fails, stopping before the first hurdle. PPO bounces energetically over each obstacle as shown in Figure 3, but sometimes falls and fails to complete the race, with an average success rate of just over 50% and an average distance of less than 110m. PULSE facilitates natural clearance of hurdles, and completes races in 17.76 seconds, a competitive time compared to human standards. Javelin throwing poses similar challenges: PPO uses inhuman strategies, AMP struggles with balancing rewards, and PULSE adopts human-like strategies but lacks specific skills for record-setting performance.

**Sports With Video Data**. For sports including golf, tennis, table tennis, and soccer penalty kick, we utilize processed motion from videos as demonstrations for AMP and PULSE+AMP. The results are reported in Table 2 and Fig. 4. In tennis, AMP demonstrates superior performance in terms of average hits; however, returned balls often land far from the intended targets. This is because prolonged rallies increase discriminator rewards, leading AMP to ignore task rewards. Notably, AMP exhibits inhuman motions at the moment of ball contact and reverts to natural movements when preparing for the next hit as shown in Fig. 4. This behavior underscores a reward conflict between balancing task and discriminator rewards. PPO plays tennis in an unnatural way, while PULSE and PULSE + AMP show similar performance. In table tennis, PPO achieves impressive error distances, but struggles with consistency and often fails to return second shots. We observe video data proves *particularly beneficial for table tennis*. PULSE+AMP records significantly higher hit averages with reasonable error distances. Table tennis requires quick reactions within a short time, which the pre-trained PULSE model supports by providing necessary motor skills, enhanced by video data that guide the learning of proper stroke techniques. For golf, penalty kicks, and free throws, the "initiating contact with an object" part makes them challenging. Here, only PULSE and PULSE+AMP manage to solve the three tasks effectively, leveraging PULSE's latent space for effective exploration. The design of these tasks often results in a sparse exploration phase where triggering penalty rewards, such as $c_t^{\text{no-dribble}}$ for moving past the ball's initial position. The AMP reward also negatively affects training penalty kick, as the human demonstration contains other soccer motions such as running and dribbling, and the policy finds them easier to learn and exploit.

**Curriculum learning**. We find curriculum learning is an essential component in achieving better results for some tasks. In Table 3, we study variants of high jump and hurdling task with and without

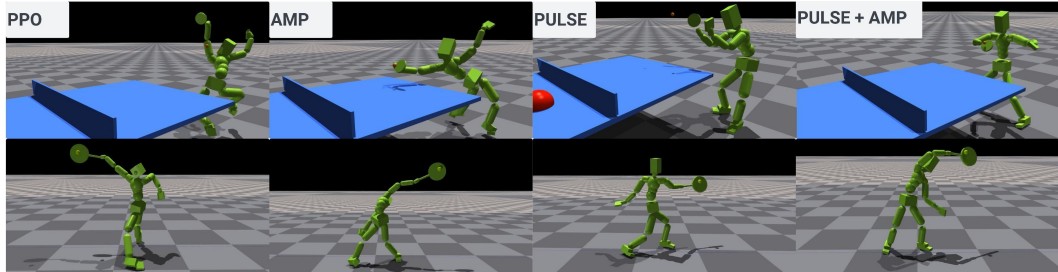

Figure 4: Qualitative results for table tennis and tennis. PPO and AMP result in inhuman behavior; PULSE can use human-like movement but PULSE + AMP result in behavior specific to the sport.

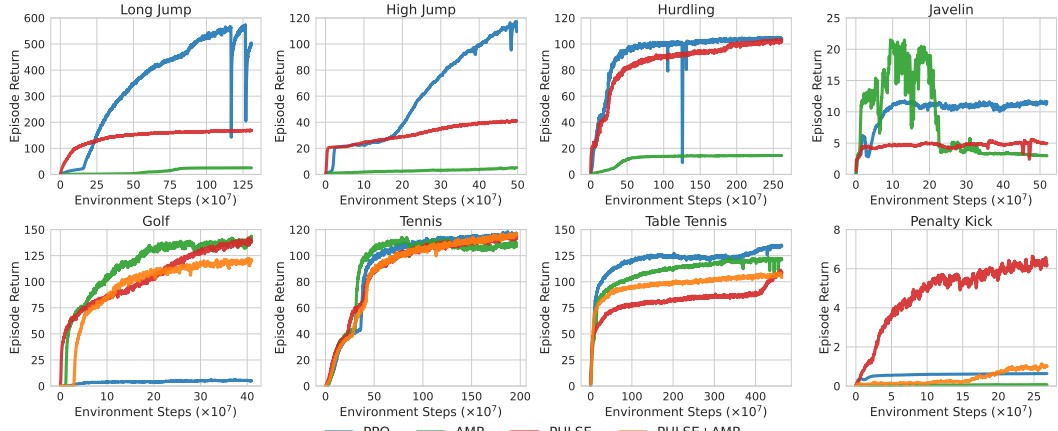

Figure 5: Learning curves on various tasks.

Table 3: Evaluation on curriculum learning.

| | High Jump | | Hurdling | | |
|---|---|---|---|---|---|
| Method | Suc Rate (1m) | Suc Rate (1.5m) | Suc Rate | Avg Dis | Time |
| w/o curriculum | **100%** | 0% | 0% | 13.65 | - |
| w/ curriculum | **100%** | **100%** | **100%** | **122.1** | **17.76** |

the curriculum using PULSE. We can see that without curriculum, high jump and hurdling both fail to solve the task. This is due to the policy not being able to obtain any reward facing challenging heights of bars and hurdles and the policy gets stuck in the local minima.

## 6   Limitations, Conclusion and Future Work

**Limitations** . While SMPLOlympics provides a large collection of simulated sports environments, it is far from being comprehensive. Certain sports are omitted due to simulation constraints (e.g., swimming, shooting, ice hockey, cycling) or their inherent complexity (e.g., 11-a-side soccer, equestrian events). Nevertheless, our framework is highly adaptable, allowing easy incorporation of additional sports like climbing, rugby, wrestling *etc*. Our initial design of rewards, though able to achieve sensible results, is also far from optimal. For competitive sports such as 2v2 soccer and basketball, our results also fall short of SOTA [8] which employs much more complex systems.

**Conclusion and Future Work**. We introduce SMPLOlympics, a collection of sports environments for simulated humanoids. We provide carefully designed state and reward, and benchmark humanoid control algorithms and motion priors. We find that by combining simple reward design and powerful human motion prior, one can achieve human-like behavior for solving various challenging sports. Our humanoid's compatibility with the SMPL family of models also provides an easy way to obtain additional data from video for training, which we demonstrate to be helpful in training some sports. These well-defined simulation environments could also serve as valuable platforms for frontier models [12] to gain physical understanding. We believe that SMPLOlympics provides a valuable starting point for the community to further explore physically simulated humanoids.

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
