# OpenReview forum: "SMPLOlympics: Sports Environments for Physically Simulated Humanoids"
_NeurIPS.cc/2024/Datasets_and_Benchmarks_Track — Submitted to NeurIPS 2024 Track Datasets and Benchmarks_

### Official Review · Reviewer_YQmi · 2024-07-07
**SMPLOlympics: Sports Environments for Physically Simulated Humanoids**

**Rating:** 6
**Confidence:** 2
**Correctness:** 1. Lines 110-111 are confusing, pleas…

**Review:**

This paper is overall well-structured, clearly written. However, there are still some issues, including several key problems that need to be addressed. Details in the `Strengths` and `Opportunities For Improvement`.

**Strengths:**

1. This paper establishes a general benchmark that supports various competitive sports tasks.
2. The baseline methods they provided in the paper is simple and effective.
3. The authors provided comprehensive and supplementary materials.

**Additional Feedback:**

This work shows interesting results, the main areas needing improvement is detailed in `Opportunities for Improvement`.

**Clarity:**

The overall paper is well written, but it also have aspects can be improved as described in `Correctness` and `Opportunities For Improvement`.

**Documentation:**

The descriptions of data collection and organization is limited in some aspect.
I can not find the URL access to the dataset.

**Limitations:**

The descriptions of the dataset are limited. Sports motion are highly challenging, and even the most advanced video-based motion capture algorithms cannot ensure the accuracy of the estimated motions. The authors extracted humanoid motions from videos but did not demonstrate the quality of the motion in the dataset.
Additionally, the ablation experiments are somewhat insufficient. Compared to previous work, the authors claim that the main advantage of this paper is the support for multiple sport tasks in one environment. What is the core advantage of this? For example, can the prior learned from a tennis task help improve performance in table tennis or even boxing?

**Opportunities For Improvement:**

The main contribution of this work is the dataset, but the authors did not provide detailed descriptions of the dataset acquisition methods and specific parameters. This undermines the main contribution of the work. Considering these issues are improvable, I gave it a score of 6.
However, if the authors do not adequately address these issues in their response, I will be inclined to consider lowering the score.

1. Are all motion in the dataset retargeted to a standard humanoid compatible with the SMPL model? Is the bone length of this humanoid constant?
2. If the bone length of the humanoid is fixed, does the error introduced by retargeting limit the realism of the dataset actions? Specifically, for multi-person interaction data, how is the accuracy of interactions ensured after retargeting?
3. In Line 44, the authors mention that tasks involving finger movements use SMPLX, while others use SMPL. Please clarify which tasks are expressed using SMPLX. There are some minor differences between the standard shapes of SMPL and SMPLX.
4. Does the Latent Motion Space for pre-training motion datasets only include AMASS? Are the newly collected sports action data used for pre-training the Latent Motion Space? Additionally, since the proposed dataset includes both SMPL and SMPLX formats, can the Latent Motion Space accommodate both SMPL and SMPLX formatted actions?
5. What are the sources of the original videos for the dataset? How many hours of data are there for each task? What are the advantages and disadvantages compared to existing datasets? It would be best to provide a table clearly explaining the dataset parameters.
6. What are the potential application scenarios for this work?
7. There is a lack of description regarding model size, the hardware environment and time required for training and testing.

**Relation To Prior Work:**

Yes, this work reasonably builds upon previous research and provides clear citations.

**Summary And Contributions:**

This work extracts competitive sports actions from videos and converts them into humanoid motion. The authors collected a dataset containing various different sports, represented uniformly as humanoid motion. For different tasks, the authors implemented baseline results using PULSE and AMP.

---

> ### Author Rebuttal · Authors · 2024-08-15
>
> Thank you for your positive score and feedback. To address your questions:
>
> ---
>
> > **Dataset Contribution**
>
> We'd like to clarify that we do not propose a dataset; our main contribution is the introduction of challenging Olympic sports environments in simulation, which serves as a new benchmark for reinforcement learning, motion synthesis, and human behavior modeling. Our data collection pipeline is primarily used to demonstrate the importance of having a small amount of task-specific data.
>
> - Q1: No retargeting is necessary as the MoCap dataset [1] and motion data extracted from videos [2] are already in SMPL format. Our humanoid's bone length is fixed, based on the neutral SMPL model with the mean body shape. Our humanoid can directly use the joint angles from the SMPL formatted motion following prior art [9]. Specific body-shaped humanoids can be constructed [9] and supported, but to use pretrained motion priors [3] trained on the mean-body-shaped humanoid, we opt to use the same setting.
> - Q2: As discriminative motion priors such as AMP [4] mainly use sequences of joint angles as the input for discerning real and fake samples, the body shape accuracy does not affect its effectiveness as a motion prior.
>
>     Reconstructing multi-person motion in global space is an emerging topic in pose estimation, and we do not extract multi-person interaction data from videos. We only extract the motion of a single player as human demonstrations. Extending single-player motion priors like AMP to a multi-person one is certainly an interesting future work that is now enabled by our multi-person environments.
>
> - Q3: The javelin throw (L160) and basketball (L224) tasks use the SMPL-X humanoid to accommodate human-object interactions. The neutral SMPL and SMPL-X humanoid indeed have different body shapes and kinematic structures. We train different motion priors for these two humanoids using the same pipeline.
> - Q4: Yes, the latent space models only use AMASS data. The demonstrations from videos are not used in training the motion latent space. We use the officially released PULSE models [3] for SMPL. For SMPL-X, We train a new PULSE-X model to support dexterous hands. We agree that learning a combined latent space for multiple body formats is an interesting and important avenue for future research. We will add these details to the manuscript.
> - Q5: We would like to respectfully emphasize that we do not propose a dataset or make any claim of superiority about the small amount of demonstration data we extracted from videos. We intend to show that a small amount of human demonstration from videos **can** improve model performance; for instance, the table tennis agent is able to learn natural paddle movement using the demonstration data as positive samples for AMP. The boxing agent also learns more skillful punching motions. We also empirically show that using a motion imitator to refine the extracted motion sequence (Appendix Section C) is helpful. We believe that these findings and our proposed pipeline can inspire future work to learn from video demonstrations and craft more large-scale carefully designed datasets to build human-like digital agents. In our supplementary materials, we provide the sources for the videos we use to extract human demonstrations.
>   In terms of the amount of demonstration data we extracted, we have 4 sequences for Tennis, 2 for Table Tennis, 6 for Soccer, 1 for Golf, and 10 for Free Throw. In total, we have 10k frames, around 5 minutes of demonstration.
> - Q6: Our suite of sports environments can be used to benchmark humanoid control and RL algorithms, focusing on humanoid sports and complementing existing benchmarks [6, 7, 8]. Since there is a plethora of existing knowledge on the preferred strategy for each sport, our environments can serve as a starting point to develop human-like embodied agents in solving sports tasks. The resulting motion can be used for animation [9] and gaming purposes [4]. Our team-sports environments also provide a test bed for developing multi-agent RL algorithms. We believe that our well-defined simulation environments serve an important role in developing future RL algorithms [10], and could also serve as a valuable platform for frontier models [5] to gain physical understanding.
> - Q7: We report our model size and training samples in Appendix B.3 Table 1. All our models are three-layer MLPs of units [2048, 1024, 512], trained on Nvidia 3090 GPU to collect about $n \times 10^9$ samples, which takes about a day. We will revise the manuscript to highlight this.
>
>
> > **Advantage for Unified Sports Environments**
>
> Our main contribution lies in proposing a collection of simulated sports environments that uses a unified humanoid embodiment (SMPL Humanoid). Such a benchmark offers insights into popular humanoid control and RL algorithms, where we show that simple rewards combined with a strong motion prior can lead to impressive sports feats. Compared to prior work, our environments offer a wide range of sports scenarios that require not only locomotion skills, but also manipulation, coordination, and planning. As prior work in humanoid sports all use different humanoids, we unify under one humanoid embodiment, so the same algorithm can be applied to different sports concurrently. While we leave multi-task learning for future work, we do show that a universal motion representation like PULSE can help almost all tasks learn. By creating a unified embodiment, important questions such as "testing knowledge transfer between skills", become accessible to the research community.

---

> > ### Author Rebuttal · Authors · 2024-08-15
> >
> > > **L110-111**
> >
> > In this sentence, we are referring to: for all values that come from MoCap, we use the "hat" $\hat{\cdot}$ symbol; we use normal symbols without accents for simulation values. The "hat" is for differentiating between states computed by human demonstration and states computed by simulation. We will revise this sentence to make it clearer.
> >
> >
> > > **References Fix**
> >
> > Thank you for bringing these to our attention. These references are now fixed.
> >
> >
> > > **References**
> >
> > [1] Mahmood, Naureen, et al. "AMASS: Archive of motion capture as surface shapes." Proceedings of the IEEE/CVF international conference on computer vision. 2019.
> >
> > [2] Wang, Yufu, et al. "TRAM: Global Trajectory and Motion of 3D Humans from in-the-wild Videos." arXiv preprint arXiv:2403.17346 (2024).
> >
> > [3] Luo, Zhengyi, et al. "Universal humanoid motion representations for physics-based control." International Conference on Learning Representations. 2024.
> >
> > [4] Peng, Xue Bin, et al. "Amp: Adversarial motion priors for stylized physics-based character control." ACM Transactions on Graphics (ToG) 40.4 (2021): 1-20.
> >
> > [5] Ma, Yecheng Jason, et al. "Eureka: Human-level reward design via coding large language models." International Conference on Learning Representations. 2024.
> >
> > [6] Brockman, Greg, et al. "Openai gym." arXiv preprint arXiv:1606.01540 (2016).
> >
> > [7] Tunyasuvunakool, Saran, et al. "dm_control: Software and tasks for continuous control." Software Impacts 6 (2020): 100022.
> >
> > [8] Sferrazza, Carmelo, et al. "Humanoidbench: Simulated humanoid benchmark for whole-body locomotion and manipulation." arXiv preprint arXiv:2403.10506 (2024).
> >
> > [9] Luo, Zhengyi, et al. "Perpetual humanoid control for real-time simulated avatars." Proceedings of the IEEE/CVF International Conference on Computer Vision. 2023.
> >
> > [10] Park, Younghyo, Gabriel B. Margolis, and Pulkit Agrawal. "Automatic Environment Shaping is the Next Frontier in RL." International Conference on Machine Learning. 2024.

---

> ### Author Response · Authors · 2024-08-31
> **Looking forward to your response!**
>
> Dear Reviewer YQmi,
>
> Thank you greatly for your helpful suggestions and feedback. They will certainly help improve our work.
>
> We have provided clarifications on each of your questions, specifically emphasizing the motivation and usage of our demonstration from the video pipeline. The simulated sports environment we propose can serve as the starting point for future efforts in simulated humanoid sports, and we have provided initial implementation, evaluation, and demonstration acquisition pipelines. We sincerely hope that our response has addressed some of your concerns and improved your impression of our work.
>
> As the author's discussion period draws near an end, please do not hesitate if there are any more questions/concerns you would like us to address.
>
> Thanks again!
>
> Authors.

---

> > ### Comment · Reviewer_YQmi · 2024-08-31
> > **Thanks for the authors.**
> >
> > Dear authors,
> >
> > When I first reviewed this paper, I hesitated between 5 and 6 points because my main concern about the data collection. After your explanation, I have a better understanding of the main contributions and techniques of this work. I appreciate the author's efforts and the many task they designed, although this may led to the author compressing the description of the backgrou and method within the limited space, but I still think this is a work worthy of acceptance, so I keep the score at 6.
> >
> > There are some small suggestions to improve the readability of this paper. Is the motion prior in Figure 2 represent the “Acquired Human Demonstration From Videos” in Sec 4.3? However, I can't find how these motion priors specifically help the model and the design of the reward function in this paper. I guess that this may be the basis of reinforcement learning for motion simulation, using methods like imitation learning. But it is better to add the reward function of motion prior to improve readability and friendliness to readers in a wider range of fields.
> >
> > Best,
> > The Reviewer

---

> > > ### Author Response · Authors · 2024-09-01
> > > **Thanks for your response!**
> > >
> > > Dear Reviewer YQmi,
> > >
> > > Thank you for your response and your acceptance recommendation!
> > >
> > > The motion prior in Figure 2 represents both the prior from “Acquired Human Demonstration From Videos” in Sec 4.3 (via AMP, adversarial motion prior) and "Large scale MoCap Data" (via PULSE). AMP serves as a motion prior via an added reward term computed by a discriminator; PULSE, on the other hand, learns a motion latent space that provides a prior action distribution. We will further clarify these two ways we use human motion data as motion prior.
> > >
> > > Thanks again!
> > >
> > > Best,
> > >
> > > Authors.

---

### Official Review · Reviewer_LLTZ · 2024-07-13
**Review on Submission78**

**Rating:** 6
**Confidence:** 3
**Correctness:** The claims appear to be correct.
**Clarity:** The paper is mostly clear with some d…

**Review:**

Pros:

- A unified sports simulation framework is interesting and helpful to the community. The efforts paid are appreciatable.
- Comprehensive experiments validate the claims well with impressive demonstrations. The comparison between PPO, AMP, and PULSE is interesting.

Questions:

- Are the human demonstrations only used for AMP-related baselines?
- How many human demonstrations are obtained for sports with video data?
- In the 2v2 soccer demonstration, the characters tend to clutter. Is there any analysis on this?

**Strengths:**

Please refer to the Review section.

**Additional Feedback:**

N/A

**Documentation:**

Documentation is sufficient.

**Limitations:**

Limitations are discussed.

**Opportunities For Improvement:**

- Including quantitative metrics for competitive sports like fencing and boxing would be preferable, e.g., ELO rating for the different baselines.
- Extensive ablation studies on the significance of human demonstrations would be helpful, like the number of demonstrations and the relevance of demonstration contents besides the demonstration quality in supplementary materials.

**Relation To Prior Work:**

Relation to prior work is discussed.

**Summary And Contributions:**

The authors proposed SMPLOlympics as an ensemble of sports simulation environments compatible with human demonstrations in SMPL(X). Experiments show that strong motion priors with simple rewards could provide impressive performance across various sports.

---

> ### Author Rebuttal · Authors · 2024-08-15
>
> Thank you for the positive rating and helpful suggestions. To address your concerns and questions:
>
> ---
>
> > **Human Demonstration Usage**
>
> The human demonstrations extracted from videos are only used for AMP-related baselines. For PULSE, we use the officially released models released by the authors, which are trained on AMASS [2] (~40 hours of MoCap data).
>
> The human demonstration data we use can be divided into two categories: tasks-specific demonstration (like the ones we obtain from videos) and general-purpose motion data (like the ones in AMASS). Our SMPL and SMPL-X compatible humanoid design can leverage both, and our results show combining a general-purpose motion prior trained on a large amount of human motion (like PULSE) with a **small amount** of **task-specific** demonstration data leads to the best performance. The general-purpose motion prior helps guide the RL training process, and task-specific data can provide specific "what skill to use" suggestions to the humanoid. Our experiments comparing PULSE [3] and PULSE + AMP [4] show the benefit of task-specific demonstration, which is especially evident for the table tennis task.
>
> For human demonstration from videos, we have 4 sequences for Tennis, 2 for Table Tennis, 6 for Soccer, 1 for Golf, and 10 for Free Throw. In total, we have around 10k frames and 5 minutes of demonstration.
>
> We will include this additional discussion.
>
> > **2v2 Soccer**
>
> For 2v2 soccer, the reward we use encourages the humanoid to chase and kick the ball toward the goal. Each humanoid has the same objective and aims to run toward the ball and kick it, and the policy has not learned to factor in its opponent's behavior. This simple formulation can serve as a starting point for more complex systems that can coordinate multi-agent play [1] and learn more interesting strategies. Notably, we believe that a motion prior like PULSE has the motor skills for dribbling, passing, and getting up from the ground, but the learning algorithm and reward formulation lack the proper expressiveness in learning such behavior.
>
> > **Fencing and Boxing Quantitative Metrics**
>
> Excellent advice! We will investigate incorporating more quantitative metrics for competitive sports. Right now, fencing and boxing have a point-based metric for hitting the opponent. Quantitatively, the agent that has been trained more recently (during competitive self-play) often dominates and obtains more points. For now, we mainly evaluate our competitive sports qualitatively, and we provide samples of fencing and boxing on our supplement site.
>
> > **References**
>
> [1] Liu, Siqi, et al. "From motor control to team play in simulated humanoid football." Science Robotics 7.69 (2022): eabo0235.
>
> [2] Mahmood, Naureen, et al. "AMASS: Archive of motion capture as surface shapes." Proceedings of the IEEE/CVF international conference on computer vision. 2019.
>
> [3] Luo, Zhengyi, et al. "Universal humanoid motion representations for physics-based control." International Conference on Learning Representations. 2024.
>
> [4] Peng, Xue Bin, et al. "Amp: Adversarial motion priors for stylized physics-based character control." ACM Transactions on Graphics (ToG) 40.4 (2021): 1-20.

---

> ### Author Response · Authors · 2024-08-31
> **Looking forward to your response!**
>
> Dear Reviewer LLTZ,
>
> Thank you very much for your valuable suggestions and feedback. They will certainly help improve our work.
>
> We have provided clarifications on how we use human demonstration data and why using the SMPL humanoid greatly improves the usability of human demonstration data in simulation. Further explanation about the soccer results is also now included in the manuscript. We sincerely hope that our response has addressed some of your concerns and improved your impression of our work.
>
> As the author's discussion period draws near an end, we would greatly appreciate it if you could acknowledge/update your initial reviews, and please do not hesitate if there are any more questions/concerns you would like us to address.
>
> Thanks again!
>
> Authors.

---

### Official Review · Reviewer_whY3 · 2024-07-26
**Initial review for SMPLOlympics**

**Rating:** 6
**Confidence:** 3
**Clarity:** Yes, the paper is well written. Secti…

**Review:**

I think the overall technical contribution may be weak. However, I believe that the unified benchmark for physically simulated humanoids is helpful for the community. Thus I rate for marginally above acceptance threshold initially.

**Strengths:**

1. The physical simulator for single person and human-human interactions of Olympic sports is significant. The humanoid which is compatible with SMPL and SMPL-X human models is also a technical contribution.
2. The benchmark for evaluating different types of Olympic sports in a standardized way with different policies is beneficial for the community.
3. The demonstrations show realistic motion results for various sports, especially for highly dynamic movements and human-object interactions.

**Additional Feedback:**

No additional feedback.

**Correctness:**

Yes, the claims are correct. The evaluation methods and experiment design are appropriate.

**Documentation:**

Yes, details are provided for reproducibility.

**Ethics:**

No ethics concerns.

**Limitations:**

Yes, the limitations and negative societal impacts are discussed.

**Opportunities For Improvement:**

1. The technical contribution is a little bit limited for that no novel policy is proposed.
2. For acquiring human demonstration from videos, are the object trajectories involved? From the demo videos, it seems that the objects are not included.
3. I think it is unnecessary to introduce every sport formulation in detail in Section 4 since they include some redundant parts;

**Relation To Prior Work:**

Yes, prior works are clearly discussed.

**Summary And Contributions:**

This paper presents a benchmark work with collection of physically simulated environments for humanoids to complete various Olympic sports. To leverage large-scale motion data from video or motion capture, the humanoids are designed to be compatible with the SMPL or SMPL-X human models. The physical simulator supports a suite of individual sports environments and competitive sports.

---

> ### Author Rebuttal · Authors · 2024-08-15
>
> Thank you for your positive feedback and constructive suggestions. To address your questions:
>
> ---
>
> > **Technical Contribution**
>
> We acknowledge that our work doesn't propose a novel policy. We focus on introducing challenging Olympic sports environments, which serve as new benchmarks for reinforcement learning and motion synthesis. We believe that these environments not only benchmark existing state-of-the-art policies but also provide a foundation for developing novel approaches to solving these challenging tasks. Also, our results on hurdling, high jump, golfing, and javelin are all **novel results** not demonstrated before in the literature. Our learned policy discovers the human-like hurdling and high jump (Fosbury flop) techniques without rewards that encourage these techniques.
>
> > **Object Trajectories**
>
> For the demonstrations collected from videos, we only record the human performance and omit the object trajectories. In addition to the challenges in estimating the object trajectory for high-speed moving objects like tennis and soccer balls, the dynamics of the simulated objects would be different from the real-world counterparts, making it difficult to completely replicate the trajectories of these objects in the simulation. Learning these interactions has been approached by specially designed imitation learning techniques [1] using graph-based rewards instead of simple task rewards. We agree that integrating reference object motion data could enhance the realism and capabilities of our learned controllers. We will further discuss this in the revised manuscript.
>
>
> > **Sports Formulation**
>
> Thanks for the suggestion! We include the state design and reward design in our manuscript for ease of access and analysis for the community. We will further shrink down the redundant parts and relocate them to the supplement.
>
> > **References**
>
> [1] Wang, Yinhuai et al. "PhysHOI: Physics-Based Imitation of Dynamic Human-Object Interaction"

---

> ### Author Response · Authors · 2024-08-31
> **Looking forward to your response!**
>
> Dear Reviewer whY3,
>
> Thank you so much for your positive comments and feedback. They will certainly help improve our work.
>
> We have provided clarifications on our technical contribution: we develop a suite of simulated sports environments that serves as benchmark and test ground for RL, animation, and human behavior modeling. We also provide novel results by leveraging existing techniques in these new environments. We sincerely hope that our response has addressed some of your concerns and improved your impression of our work.
>
> As the author's discussion period draws near an end, we would greatly appreciate it if you could acknowledge/update your initial reviews, and please do not hesitate if there are any more questions/concerns you would like us to address.
>
> Thanks again!
>
> Authors.

---

### Official Review · Reviewer_SaX1 · 2024-08-05
**SMPLOlympics review**

**Rating:** 6
**Confidence:** 3
**Correctness:** Seems correct
**Clarity:** Well written.

**Review:**

See below.

**Strengths:**

1. This paper meticulously constructs multiple Olympic sports simulations by formulating each value function.
2. This paper demonstrates that extracting data from videos and incorporating it into the training process results in significantly improved learning performance compared to models without data injection.

**Additional Feedback:**

NA

**Documentation:**

Well documented.

**Limitations:**

See above

**Opportunities For Improvement:**

1. The model presented in this paper lacks realism. It would be beneficial to consider joint position and velocity limitations, as well as other real motor constraints, to ensure practical applicability to physical robots. Without accounting for these factors, the trained model cannot be effectively implemented on actual humanoid robots.
2. Regarding the reward function, it is recommended that the authors provide a detailed analysis for each component. For instance, plotting the progression of each component as the learning epochs increase would illustrate the necessity of each element. Without such an ablation study, it is challenging to demonstrate the effectiveness of the reward function.
3. The joint actuation forces are set to 500 Nm, which is unrealistic given the capabilities of state-of-the-art humanoid robots.

**Relation To Prior Work:**

Clearly discussed.

**Summary And Contributions:**

This paper introduces a comprehensive framework for simulating multiple Olympic sports within a virtual environment for humanoid robots. The dataset for these simulations is derived from video recordings and motion capture (moCap) technology. A variety of reward structures have been designed to facilitate the learning process across different sports disciplines, including high jump, hurdling, golf, and javelin. In addition to individual sports, the paper also explores team and multi-person sports such as table tennis, fencing, boxing, soccer, basketball, and competitive self-play scenarios.

The study employs various learning-based models to compare performance, specifically focusing on Proximal Policy Optimization (PPO), Adversarial Motion Prior (AMP), Predictive Uncertainty for Learning and Safe Exploration (PULSE), and a combined PULSE+AMP approach. Experimental results demonstrate that the inclusion of reference data significantly enhances the learning efficiency of the humanoid robots, enabling them to perform sports activities with greater proficiency.

---

> ### Author Rebuttal · Authors · 2024-08-15
>
> Thank you for your positive comments and constructive feedback. To address your concerns:
>
> ---
>
> > **Humanoid Realism**
>
> We agree that incorporating realistic constraints, like real-world humanoids (e.g., Unitree H1), is a promising future direction. The current humanoid is built for efficient learning and recreating the agile motor skills seen in human sports, and we plan to support more realistic humanoids in the future. Our environments and reward designs can be adapted to such efforts. We believe it is beneficial to first tackle the simpler tasks and study humanoid behavior in simulation before tackling them in the real world.
>
> Regarding joint limits, our joint limit design is mostly to accommodate the available data in SMPL format. The MoCap dataset (AMASS [1]) and pose estimation methods from the vision community usually do not consider joint limits. As a result, introducing joint limits on our humanoid could lead to unnatural motion when imitating the input motion. More realistic joint limits will certainly be introduced once we move toward more realistic humanoids.
>
>
> > **Reward Function Analysis**
>
> We will expand our manuscript with component-wise reward analysis and discussion on the relationship between reward design and success rate. While we report reward curves, the main focus for evaluation is the task success rates, each sport's respective metrics, and the resulting humanoid behavior. Our current reward design, while effective, serves as a basis for further improvement.
>
>
> > **Joint Actuation Forces**
>
> We acknowledge that the 500Nm joint torque limit is high for current humanoid capabilities. It does roughly match existing robots when normalized to the robot's weight. Comparing our simulated humanoid to the Unitree H1:
>
> Unitree H1:
> - Weight: 47kg
> - Maximum joint torque: 360Nm
> - Normalized torque-to-weight ratio: 7.6Nm/kg
>
> Our simulated humanoid:
> - Weight: 70kg
> - Maximum joint torque: 500Nm
> - Normalized torque-to-weight ratio: 7.1Nm/kg
>
> While our simulated motors don't exactly match realistic motors, the normalized torque-to-weight ratios are comparable. Our simulated environments and humanoid setup allow us to explore complex behaviors in simulation while using similar joint torques to prior work [2], and could facilitate potential future transfer to real-world applications.
>
> > **References**
>
>  [1] Mahmood, Naureen, et al. "AMASS: Archive of motion capture as surface shapes." Proceedings of the IEEE/CVF international conference on computer vision. 2019.
>
>  [2] Peng, Xue Bin, et al. "Amp: Adversarial motion priors for stylized physics-based character control." ACM Transactions on Graphics (ToG) 40.4 (2021): 1-20.

---

> ### Author Response · Authors · 2024-08-31
> **Looking forward to your response!**
>
> Dear Reviewer SaX1,
>
> Thank you so much for your constructive feedback and suggestions. They will certainly help improve our work.
>
> We have provided comments to your concerns on humanoid realism and updated our paper to include additional analysis and discussion on reward and sim-to-real. We sincerely hope that our response has addressed some of your concerns and improved your impression of our work.
>
> As the author's discussion period draws near an end, we would greatly appreciate it if you could acknowledge/update your initial reviews, and please do not hesitate if there are any more questions/concerns you would like us to address.
>
> Thanks again!
>
> Authors.

---

### Author Rebuttal · Authors · 2024-08-15

# General Response

We thank the reviewers for their time and constructive feedback. We are glad that they find our environments "meticulously constructed" (SaX1), "significant" (whY3), "beneficial to the community" (whY3, LLTZ), our simulation result "impressive demonstrations" (LLTZ), and details "comprehensive" (YQmi). Here, we answer some common questions and provide a list of revisions made.

**Demonstration Data**

Our main contributions are a collection of simulated sports environments that uses a unified humanoid embodiment (SMPL(X) humanoid),  their accompanying baseline state & reward design, and benchmarking state-of-the-art character control algorithms. Our SMPL-compatible embodiment enables directly using human demonstration data, either from MoCap or videos. We show the importance of using demonstration data in two ways: (1) we show that PULSE, a motion representation/prior trained on large-scale MoCap data can help learn many sports tasks with simple reward designs and result in human-like behavior; (2) we propose a pipeline to obtain a **small amount** of **task-specific** demonstration from video for each sport (when there is no available public dataset), and show that it can further help shaping the humanoid behavior when used together with a general-purpose motion prior. We will provide the tasks-specific motion sequences (extracted from videos) we use for future research, but we do not claim a proposed dataset or its superiority over prior datasets.


Based on the feedback, we have made the following revisions:

- Component-wise reward progression analysis.
- More details about obtaining demonstrations from videos, including the number of sequences and the clear purpose of this proposed pipeline.
- Discussion on sim-to-real possibility and joint realism.
- Removing redundant information about each sports formulation.

---

### Decision · Program_Chairs · 2024-09-26

**Decision:**

Reject

**Comment:**

Note from PC: This year, the track has been incredibly competitive, which meant that many good papers had to be rejected. After careful discussion with the SACs we have concluded that this paper unfortunately cannot be accepted this time. This is the final decision, which cannot be appealed. We encourage the authors to incorporate feedback from reviewers and additional results / discussion provided during the author response period in their next submission.

This work presents a benchmark work with a collection of physically simulated environments for humanoids to complete various Olympic Sports. All reviewers consistently recommended accepting this work. AC agrees that this work is interesting and deserves to be published on the NeurIPS dataset track 2024. The reviewers did raise some valuable concerns that should be addressed in the final camera-ready version of the paper. The authors are encouraged to make the necessary changes in the final version.